# Antibacterial Activity of Aureonuclemycin Produced by *Streptomyces aureus* Strain SPRI-371

**DOI:** 10.3390/molecules27155041

**Published:** 2022-08-08

**Authors:** Weiguo Wang, Minkang Feng, Xiaomeng Li, Feiyu Chen, Zhihao Zhang, Wenlong Yang, Chen Shao, Liming Tao, Yang Zhang

**Affiliations:** 1Shanghai Key Laboratory of Chemical Biology, School of Pharmacy, East China University of Science and Technology, Shanghai 200237, China; 2Shanghai Jiading District Agricultural Machinery Technology Promotion Station, Shanghai 201800, China; 3School of Bioengineering, East China University of Science and Technology, Shanghai 200237, China

**Keywords:** aureonuclemycin, *Xanthomonas oryzae* pv. *oryzae*, *Xanthomonas citri* subsp. *citri*, *Xanthomonas oryzae* pv. *oryzicola*, antibacterial activity

## Abstract

Actinomycetes play a vital role as one of the most important natural resources for both pharmaceutical and agricultural applications. The actinomycete strain SPRI-371, isolated from soil collected in Jiangsu province, China, was classified as *Streptomyces aureus* based on its morphological, physiological, biochemical and molecular biological characteristics. Its bacterial activity metabolites were identified as aureonuclemycin (ANM), belonging to adenosine derivatives with the molecular formula C_16_H_19_N_5_O_9_ for ANM A and C_10_H_13_N_5_O_3_ for ANM B. Simultaneously, the industrial fermentation process of a mutated *S. aureus* strain SPRI-371 was optimized in a 20 m^3^ fermentation tank, featuring a rotation speed of 170 rpm, a pressure of 0.05 MPa, an inoculum age of 36–40 h and a dissolved oxygen level maintained at 1–30% within 40–80 h and at >60% in the later period, resulting in an ANM yield of >3700 mg/L. In the industrial separation of fermentation broth, the sulfuric acid solution was selected to adjust pH and 4# resin was used for adsorption. Then, it was resolved with 20% ethanol solution and concentrated in a vacuum (60–65 °C), with excellent results. Antibacterial experiments showed that ANM was less active or inactive against *Xanthomonas oryzae* pv. *oryzae*, *Xanthomonas citri* subsp. *citri* and *Xanthomonas oryzae* pv. *oryzicola* and most bacteria, yeast and fungi in vitro. However, in vivo experiments showed that ANM exhibited extremely significant protective and therapeutic activity against diseases caused by *X. oryzae* pv. *oryzae* and *X. oryzae* pv. *oryzicola* in rice and *X. citri* in oranges and lemons. In field trials, ANM A 150 gai/ha + ANM B 75 gai/ha exhibited excellent therapeutic activity against rice bacterial leaf blight, citrus canker and rice bacterial leaf streak. Furthermore, as the dosage and production cost of ANM are lower than those of commercial drugs, it has good application prospects.

## 1. Introduction

In modern agriculture, pesticides are frequently used to ensure an economic yield from crop plants. However, the abuse of traditional pesticides has led to problems, particularly synthetic fungicides, because of the possible adverse effects of residual chemicals on the environment [1] and the emergence of fungicide-resistant pathogens [2]. Therefore, scientists are attempting to develop modern fungicides that should be effective, environmentally friendly and economical. In this context, they have been focusing on the use of microorganism-based agents as alternatives to, or in combination with, synthetic chemical fungicides for controlling the spread and severity of plant diseases [3]. Some studies indicate that microbial metabolites may help overcome resistance and pollution problems due to their versatility in structure and enhanced biodegradability [4]. The exploration of natural actives, specifically with new chemical structures, is an important aspect, with high potential in the field of both medicine and phytomedicine [5].

Metabolites produced by microorganisms are crucial sources of antifungal compounds [6]. They may be developed as products directly or as a lead compound for chemical optimization [7,8]. In this regard, Streptomyces strains are the primary source to obtain metabolites. Although thousands of metabolites with antifungal or antibacterial activity have been extracted from various Streptomyces strains, unfortunately, most of them are not yet commercialized. The most important reasons for this issue are that the yields of metabolites in fermentation are too low to have the economic feasibility for large-scale production. The high production cost does not enable their competition with existing synthetic compounds. Hence, the first and quintessential target of all microorganism-sourced activities is increasing the yields of metabolites to an economical level. In addition to mutant strains, optimization of fermentation conditions is another efficient method to increase the yields of metabolites. As shown by studies, regulating pH levels significantly affected Aspergillus nidulans’ protein production, and increasing dissolved oxygen levels stimulated aryl alcohol oxidase production [9]. In general, the optimization of the fermentation process is primarily focused on the optimization of the culture medium and basic culture conditions in the laboratory stage, whereas in the pilot test stage, the optimization is focused on parameters, such as fermentation tank form, aeration ratio and form, sterilizing conditions and seed quality. In the final large-scale stage, some process conditions could still be optimized for the finalized production equipment. Some studies have found that the inoculum size, temperature, pH, stirring and aeration rates, dissolved oxygen tension and fermentation duration exert a certain impact on the yield in the production of bacteriocin from the Bacillaceae family [10].

*Xanthomonas oryzae* pv. *oryzae* and *Xanthomonas oryzae* pv. *oryzicola* cause rice bacterial leaf blight and rice bacterial leaf streak. These two diseases have been reported to varying degrees in rice-producing areas in Southern China and Southeast Asia, typically in humid, muggy and typhoon-prone Southeast Asia [11]. A severe infection of this disease often prevents a successful rice harvest. Similarly, *Xanthomonas citri* subsp. *citri* also causes a severe bacterial disease in citrus, occurring in citrus-producing areas across the world [12]. This disease not only affects the yield but also gravely affects the quality of citrus. Currently, biological and physical methods, such as planting resistant strains [13,14], pruning and burying diseased branches, are primarily used to control citrus canker [15]. Some agents containing copper were also primarily used as crop control measures, but they can increase potential risks to human health [16,17,18]. In addition, phytopathogenic bacterial populations develop resistance to copper by exchanging plasmids containing resistance-conferring genes [19]. These three bacterial diseases are highly contagious and there are no good prevention and control measures. Therefore, many countries and regions, including China, have listed these three diseases as quarantine objects.

This study describes the taxonomy of the strain SPRI-371, the optimization of the industrial fermentation process, the separation of metabolites and the field tests. It also demonstrates the potential application of the metabolite aureonuclemycin (ANM). The ANMs were A and B types, which were elucidated to be adenosine derivatives with the molecular formulae C_16_H_19_N_5_O_9_ and C_10_H_13_N_5_O_3_, respectively (Figure 1). The ANM A was confirmed to be a nucleoside antibiotic with adenine, similar to herbicidin [20], and ANM B possessed 5’-deoxyadenosine, exhibiting antibacterial activity.

## 2. Results

### 2.1. Identification of the Strain SPRI-371

The strain SPRI-371 grew well at pH 5–10. Growth was observed between 20 and 40 °C. The aerial hyphae indicated monopodial branching with sporophores of rectus-flexibilis on most of the media tested and spira occurred on some media. The sporophore hyphae present a 2–6 screw helix. The spores were oval in shape with a smooth surface. On most of the media, the color of the substrate mycelium was grayish to brown and the mass color of the aerial mycelia was white to grayish. Cultural characters were summarized in Table 1. The strain SPRI-371 reduced nitrate to nitrite and produced H_2_S and pigment. Casein and gelatin were hydrolyzed. There was no hydrolysis of starch or cellulose. D-glucose, L-arabinose, D-xylose, D-fructose, mannitol and L-rhamnose were well utilized; i-inositol was utilized, but not saccharose nor raffinose. Physiological properties and utilization of carbon sources are shown in Appendix A. After comparison with those of the known species of *Streptomyces*, *S. aureus* was selected as the most closely related one.

Comparing the strain SPRI-371 to *S**. aureus* SF 1836, as shown in Appendix A, the differences were evident from the data described in the literature. There were differences, including the peptonization of milk, production of H_2_S and cultural characteristics. *S. aureus* SF 1836 was the most closely related species of Staphylococcus aureus among the known Streptomyces species. Therefore, the strain SPRI-371 was identified as a variant species in *S. aureus*. In conclusion, the cultural characteristics and carbon source utilization of the strain SPRI-371 were determined, which could be used for later research on the industrial fermentation process.

Based on the 16s rRNA sequence analysis, the strain SPRI-371 seemed to be classified as *S. lanatus*, but the morphological, biochemical and physiological characteristics were quite different between the strain SPRI-371 and the typical strain *S. lanatus* ISP509015. Though the comparability of the 16S rRNA sequence between SRPI-371 and *S.*
*aureus,* AY09436816 was only 95%, and the strain SPRI-371 was still identified as a variation genus of *S. aureus* on the basis of the simulation and difference observed. The strain SPRI-371 was also shown in the neighbor-joining tree (Appendix A), close to *S. lanatus* and *S. aureus*.

### 2.2. Results of Fermentation Process Optimization

Under the condition that other parameters remain unchanged, the speed of 170 rpm was the most suitable and the yield of ANM could reach more than 3700 mg/L (Figure 2A). The result indicated that 0.05 MPa was the most suitable for the internal pressure of the tank when other parameters remain the same and the yield of ANM could reach more than 3900 mg/L (Figure 2B).

Inoculum quality had a great impact on fermentation, as Figure 2C illustrates. If the inoculum age was too short, the fermentation period should be prolonged and the yield was not high. Too-long an inoculum age, mycelium aging, is not conducive to fermentation. According to the microscopic observation of the mycelium production of the inoculum, the yield reached the highest when the inoculum grew to the mycelium level and a few vacuoles appeared. Therefore, the yield was best when the inoculum age was about 36 h.

In the fermentation process, dissolved oxygen was an important factor and the change in dissolved oxygen was regulated by treating the ventilation rate and stirring speed. To this end, we designed different dissolved oxygen states in a different way, as described in Appendix A, and observed the influence on fermentation. It could be seen from Figure 2D that when the tank was in a low-dissolved oxygen state for a long time in the fermentation process, the yield decreased obviously. The production was higher than that of low-dissolved oxygen but worse than that of high at both ends and low in the middle. During the fermentation process, a dissolved oxygen level maintained at 1–30% within 40–80 h and at >60% in the later period resulted in an ANM yield of >3700 mg/L.

### 2.3. Results of Separation Process Optimization

The purpose of fermentation broth pretreatment was to coagulate protein, increase filtration speed and improve filtrate quality. Different solvents were used to adjust to different pH and different filtration rates for pretreatment. The experimental results are shown in Figure 3A,B. We found that when adjusted to different pipes, the fermentation broth showed different turbidity after filtration, and the filtrate showed a transparent state only when sulfuric acid was used to adjust the pH to between 1 and 4. The sulfuric acid solution was used to adjust the pH between 2 and 6 for filtration, and the recovery rate of ANM was above 98%. Considering the feasibility and cost of industrial operation, the optimal pretreatment conditions were determined. The optimal condition was to adjust the pH to 3–4 with the sulfuric acid solution before filtration and the filtration rate was 10 mL/min.

The filtrate was adsorbed and decomposed with macroporous adsorption resin. The experimental results are shown in Figure 3C,D. we found that the adsorption capacity of resins 1#–5# varies greatly and the adsorption capacity of 4# resin is the largest, which could reach 20,000 mg/mL. The collection rate after analysis was the same, among which the collection rate of 4# resin was the highest, exceeding 95%. Therefore, 4# resin was the best choice for adsorption and desorption in industrial separation.

Considering that ANM was insoluble in organic solvents, different concentrations of ammonia and ethanol solutions were used for analysis. The experimental results are shown in Figure 3E,F. We found that the analytical effect of ethanol aqueous solution was better than that of ammonia water. Among them, 20% ethanol aqueous solution had the best analysis effect and the collection rate was close to 95%.

In the presence of water, ANM becomes increasingly unstable as temperature increases. Therefore, the concentration temperature of the analytical solution directly affects the separation yield and product quality in ANM. In this experiment, the effect of concentration temperature on the collection rate was explored and the results are shown in Figure 3G. Two methods of atmospheric pressure and vacuum were used and the concentration time was different. It was found that the collection rate remained above 90% at the temperature of 40–65 °C. Taking into account the convenience and cost of operation, the concentration time at 60–65 °C was short and the collection rate could exceed 92%, which was the best choice.

After the concentration was completed, the crystallization solution of ANM could be precipitated by adjusting the pH. Then, the solution was stirred evenly and left to stand overnight. After crystallization, the vacuum-drying method was used to dry. The temperature was required to be controlled at 50–55 °C and dry for 5–6 h to meet the quality standards.

### 2.4. Evaluation of In Vivo Antibacterial Activity of ANM

#### 2.4.1. Results of ANM A on the Preventive Effect of Rice Bacterial Leaf Blight Caused by *X. oryzae* pv. *oryzae*

The in vivo efficacy of ANM A for the prevention of the leaves of rice plants from bacterial rice bacterial leaf blight caused by *X. oryzae* pv. *oryzae* was evaluated under greenhouse conditions and the result was demonstrated, as shown in Figure 4A. After 10 days of the administration, it could be seen that 188 mg/L ANM A had a good preventive effect on leaf disease, which was about twice that of 20% Bismerthiazol, and there was a significant difference. After 20 days of treatment, it could be seen that 188 mg/L ANM A, 140 mg/L ANM A and 1250 mg/L 20% Bismerthiazol had the same preventive effect and there was no significant difference. After the piercing leaf method was used to inoculate the pathogen, the condition of the leaves was observed (Figure 4C) and it was clearly seen that there was a large number of disease spots and water spots around the small holes of the leaves in the control group. However, no disease spots appeared around the leaves in the ANM-treatment group. Therefore, 10 days after the application, ANM A showed a good effect on the prevention of rice bacterial leaf blight, the effect was better than that of the commercially available Bismethiazole and the dosage was also lower.

#### 2.4.2. Results of ANM A on the Therapeutic Effect of Citrus Canker Caused by *X. citri*

The tests in Guangdong, Hunan, Fujian, Shanghai in China and Supan Province in Thailand showed that ANM A had a significant therapeutic effect on citrus canker caused by *X. citri* on various orange or lemon trees. Figure 4B shows the results obtained in the orange garden of Tianbian town, Nanhai County in Guangdong province. When the disease index was low, 112.5 mg/L ANM A had the same therapeutic effect as copper humic acid and there was no significant difference. However, the therapeutic effect of 187.5 mg/L ANM A was better than that of copper humic acid and there was a significant difference. In general, ANM A was used less than copper humic acid to achieve the same therapeutic effect.

### 2.5. Evaluation of In Vivo Antibacterial Activity of ANM (A + B)

#### 2.5.1. Results of ANM (A + B) on the Preventive and Therapeutic Effect of Rice Bacterial Leaf Blight Caused by *X. oryzae* pv. *oryzae*

The application of ANM (A + B) to prevent and therapize rice bacterial leaf blight caused by *X. oryzae* pv. *oryzae* was also tested in Hunan and Jiangxi provinces in China under field conditions. The preventive and therapeutic effect of Bis-ADTA or ANM (A + B) combined on rice bacterial leaf blight is shown in Figure 5. The preventive effect (Figure 5A) of ANM A 150 gai/ha + ANM B 75 gai/ha was the best, both exceeding 90%. The therapeutic effect (Figure 5B) of ANM A 75 gai/ha + ANM B 37.5 gai/ha was better than that of Bis-ADTA 187.5 gai/ha and there was a significant difference. In conclusion, ANM (A + B) could achieve the same control effect as Bis-ADTA at a lower dose than commercial Bis-ADTA.

#### 2.5.2. Results of ANM (A + B) on the Preventive and Therapeutic Effect of Citrus Canker Caused by *X. citri*

According to the results in Figure 5C, there was a difference in the therapeutic effect of citrus canker using Kocide or ANM (A + B). ANM A 150 gai/ha + ANM B 75 gai/ha had the best therapeutic effect and it could be seen that the therapeutic effect exceeded 85%. The therapeutic effect of ANM A 75 gai/ha + ANM B 37.5 gai/ha exceeded 80%, which was higher than that of Kocide 965.25 gai/ha, and there was a significant difference. In conclusion, the therapeutic effect of ANM (A + B) was better than that of Kocide and the dosage was lower.

#### 2.5.3. Results of ANM (A + B) on the Preventive and Therapeutic Effect of Rice Bacterial Leaf Streak Caused by *X. oryzae*

The therapeutic effect of Bismethiazole or ANM (A + B) on rice bacterial leaf streak was tested. The results are shown in Figure 5D. The therapeutic effect of ANM A 150 gai/ha + ANM B 75 gai/ha was more than 85%, while the treatment effect of Bismethiazole 300 gai/ha was about 70% and there was a significant difference between them. In conclusion, the therapeutic effect of ANM (A + B) was better than that of Bismerthiazol and a relatively ideal therapeutic effect could be achieved at a lower dose.

## 3. Discussion

The morphological, biochemical and physiological characteristics of SPRI-371 were evaluated to determine its cultural characteristics and carbon source utilization. Simultaneously, through a comparative analysis of the 16S rRNA sequence, it was found that SPRI-371 was closer to *S. lanatus*, but there were large differences in the morphological, biochemical and physiological characteristics [21,22]. Its location was shown in the neighbor-joining tree with a 99% approximation to *S. aureus*. The strain SPRI-371 was finally determined to be a mutant strain of *S. aureus*.

Stirring speed, pressure and inoculum age in industrial fermentation were the routine investigation parameters [23] and their optimal values were determined in this study. Dissolved oxygen is the most variable parameter of different strains in the fermentation process and the demand for the same strain during each period is also different. However, considering the limitation of tank structure in large-tonnage production, the coordinated control of ventilation rate, stirring speed and tank pressure could result in different dissolved oxygen values. Finally, the following conditions were determined to use SPRI-371 in large-tonnage industrial fermentation: the speed at 170 rpm, tank pressure at 0.05 MPa and inoculation time for 36 h. Moreover, dissolved oxygen levels should be maintained at 1–30% within 40–80 h and at >60% in the later period. Finally, the yield of ANM could reach >3700 mg/L.

In industrial separation, the pre-treatment of fermentation broth, the selection of adsorption resin [24] and analytical solution and the determination of concentration temperature are important factors affecting the final yield of ANM. In this study, it was discovered that when the pH was adjusted to 2–4 using sulfuric acid, the filtrate was transparent and the ANM collection rate was high. However, considering the efficiency, the final choice was to adjust the pH to 3–4 using sulfuric acid and the filtration rate was 10 mL/min for pre-treatment. For the selection of the adsorption resin and analytical solution, the amount of adsorption and the collection rate after analysis were considered. It was observed that adsorption with 4# resin and analysis using 20% ethanol aqueous solution provided the best effect. For concentration, the temperature is an important parameter [25]. Although vacuum treatment is performed at 40–60 °C and the collection rate of the concentration is the same, it is necessary to consider efficiency and cost issues. Finally, a temperature range of 60–65 °C and vacuum concentration for 4.8 h resulted in an ANM collection rate of >90%.

*S. aureus* is well known for the production of diverse bioactive compounds, such as polyalkenes [26], possessing antifungal activity, aureomycin and antimycin [27,28]. In the screening of new microbial pesticides, ANM A and ANM B were isolated from the *S. aureus* strain SPRI-371 and some plant pathogens were found to exhibit strong antibacterial activity in vivo. The antibiotic ANM was isolated from the culture filtrate of *S. aureus* SPRI-371 using various chromatographic procedures. The ANM A was confirmed to be a nucleoside antibiotic with adenine and analogous to herbicidin [29]. The only difference between ANM A and herbicidin C was that at the C8 position, ANM A had a carboxyl group, and herbicidin C had a methyl ester group. Herbicidin has been reported to exhibit high herbicidal and antibacterial activities in vivo, even against Cryptosporidium parvum [29]. Nucleoside antibiotics, such as blasticidin S and polymyxin [30], exhibit various biological activities, such as antibacterial, antifungal, anticancer and herbicidal. In some studies, sulfa-ethyl amide was introduced into the purine nucleoside structure and a series of new sulfa-containing purine nucleoside derivatives were designed and synthesized, which were found to be effective against tobacco mosaic virus, cucumber mosaic virus and potato virus Y [31]. The structure of ANM B belongs to 5′-deoxyadenosine. Studies demonstrated that 5′-deoxyadenosine exhibits certain activities [32], including activity against parasites [33]. Furthermore, deoxyadenosine reverses the hydroxyurea inhibition of vaccinia virus growth [34] and exhibits anti-HIV activity [35]. Both ANM A and ANM B are nucleoside derivatives exhibiting potential antibacterial activity. In addition, bacteria, being lower organisms, sometimes use simple structures in the environment when synthesizing nucleotides, resulting in replication misplacement or fragmentation, which may be the potential antibacterial mechanism of nucleoside derivatives.

In the in vivo antibacterial experiments, it was found that a single ANM A molecule exerted an excellent therapeutic effect on rice bacterial leaf blight and citrus canker and the combined use of ANM (A + B) exerted a better effect. Therefore, in the follow-up study, only the therapeutic effect of the ANM combination (A + B) was investigated related to measurements. Experiments revealed that ANM A 150 gai/ha + ANM B 75 gai/ha could effectively treat rice bacterial leaf blight, citrus canker and rice bacterial leaf streak, with an average treatment effect of >85%. It is worth mentioning that the effect of ANM (A + B) was even better than that of commercial pesticides and the dosage was lower than that of commercial agents. The therapeutic effect of ANM A 75 gai/ha + ANM B 37.5 gai/ha was also better than that of commercial pesticides and the dosage was lower. After dosage calculation, it was found that when the ANM combination (A + B) was used, the dosage per hectare was 45–300 g, whereas it is 300–1000 g for other commercially available pesticides. During actual production, agricultural and sideline products, such as soybean meal, corn meal and feather meal, could be completely utilized for industrial fermentation, thereby reducing costs. Therefore, in practical production applications, the production cost of ANM is lower. As a metabolite of *S. aureus* SPRI-371, ANM exhibits good stability, safety (acute toxicity in mice: LD_50_ > 5000 mg/kg body weight) and low toxicity; moreover, it could effectively control rice bacterial leaf blight, citrus canker and rice bacterial leaf streak. Therefore, the ANM combination (A + B) is an ideal substitute for pesticides available on the market because of its low industrial production cost and good therapeutic effect.

## 4. Materials and Methods

### 4.1. Materials

All chemical reagents for laboratory use were purchased from Aladdin, biochemical reagents were purchased from Macklin, Shanghai, China, and materials for the industrial test were obtained from the market.

### 4.2. Antibiotic-Producing Organism the Strain SPRI-371

The strain SPRI-371 was isolated from the soil in Jiangsu Province in China and investigated for the production of an antibacterial substance. It was routinely cultured on GS agar at 28 ± 1 °C and preserved in the dry milk at −70 °C for long-term maintenance.

### 4.3. Taxonomic Studies

The culture characters of the microorganism, which produce ANM, were determined by the use of the media and methods described by Shirling and Gottlieb [36]. Observations were made after the cultures were incubated at 28 ± 1 °C for two weeks, except where otherwise mentioned. The taxonomic key of Bergey’s Manual (9th ed) and of Waksman in the actinomycetes [37] were used to compare cultures with recognized genera and species of the actinomycetes.

The 16S ribosomal RNA sequence analyses were performed to determine its classification. Methods for the separation of total DNA were described in the Laboratory Manual of Genetic Manipulation of *Streptomyces*. The primers of PCR were 5′-AGAGTTTGATCCTGGCTCAG-3′ and 5′-AAGGAGGTGATCCAGCCGCA-3′ (Designed by Tsingke Biotechnology Co., Ltd., Beijing, China). Then it was heated to 95 °C for 1 min, cooled to 55 °C for 1 min, delayed to 72 °C for 3 min and the cycle was repeated 30 times. TG1 was used as a receptor and pMD18-T as a vector. Plasma DNA was confirmed by electrophoresis and analysis by PCR.

### 4.4. Culture Conditions for Antibiotic-Producing

*S. aureus* strain SPRI-371 was precultured in a 250 mL flask containing 50 mL medium at 32 °C and with shaking at 210 revs ·min^−1^ for 48 h. The medium contained glucose 2%, soybean power 2%, peptone 0.1%, NH_4_NO_3_ 0.6%, NaCl 0.2%, KH_2_PO_4_ 0.005%, CaCO_3_ 0.6% and pH was adjusted to 7.2 before sterilization. This precultured broth was used to inoculate to 14 L auto fermentor with 10 L medium at 32 °C for 90 h. The medium for the production contained glucose 5%, starch 1%, soybean power 1%, peanut power 1%, NH_4_NO_3_ 0.6%, NaCl 0.2%, KH_2_PO_4_ 0.005%, CaCO_3_ 0.6% and pH was adjusted to 7.2 before sterilization.

### 4.5. Isolation and Purification of Antibiotic Substances in Laboratory

The cultured broth (10 L) was filtrated to remove the cells and the filtrates were passed through a pretreated column with Diaion HP-20 (1 L). After rinsing with water (3 L), the column was eluted with stepwise gradients of ethanol + water (0 + 100, 20 + 80, 40 + 60, 60 + 40, 80 + 20 and 100 + 0 by volume). Each fraction (2 L) of the eluates was concentrated under reduced pressure using a rotary evaporator. The active fraction was concentrated to dry (about 24 g). The sample was then extracted with 1 L methanol to remove MeOH-insoluble impurities and concentrated to yield a crude crystal (22 g). Further purification was carried out by chromatography on a silica gel column. The adsorbate was analyzed by HPLC (C18 retention phase, 260 nm) (Agilent, Shanghai, China), eluted with a solvent system consisting of different ratios of CH_3_Cl_3_ and 500 mL fractions were collected. The fraction was concentrated to dryness (21.5 g) in a vacuum. The dryness was then transferred to hot water (1 L) and the pH was adjusted to 3.8. Next, it was cooled under room temperature and about 16 g crystal (98%) occurred. Finally, recrystallization was performed to obtain pure crystals (99.2%).

### 4.6. ANM A and ANM B Produced by Industrial Methods

The matured *S. aureus* strain SPRI-371 spores were inoculated in the sterilized inoculum ling medium consisting of glucose 2%, starch 2%, soybean power 2%, NaCl 0.2%, NH_4_NO_3_ 0.4%, KH_2_PO_4_ 0.05% and cultured at 28 °C for 36–40 h. The culture was inoculated in a fermentation medium consisting of glucose 2%, starch 5%, peanut cake powder 1%, soybean cake powder 2%, NaCl 0.2%, NH_4_NO_3_ 0.4%, and CaCO_3_ 2% at a ratio of 5–20% and incubated at 32 °C for 90–96 h. Further, 100 kg of fermentation broth was taken and filtered by the membrane to obtain about 80 kg of filtrate. The filtrate pH was adjusted to 3–4 with 2 mol hydrochloric acid and then adsorbed on macroporous adsorption resin (1300 type, commercially available). After washing with water, stepwise desorption was carried out using 5 kg of 5% acetone to get ANM B and 5 kg of 30% acetone to get ANM A, respectively. The first desorption solution was concentrated to obtain 200 g of yellowish ANM B crystals (containing about 90% of ANM B). About 180 g of purified ANM B (94%) was obtained after recrystallization. The second desorption solution was concentrated and 350 g of yellowish ANM A crystals (about 85% ANM A) were obtained. After recrystallization, about 300 g of colorless needle-shaped ANM A crystal was obtained (about 95% of ANM A).

### 4.7. Optimization of the Fermentation Process

According to the structure difference between the pilot test scale and the large fermentation tank, 20 tons fermentation tank was mainly used to optimize the process parameters. Parameters included stirring speed, pressure, inoculum age and dissolved oxygen.

The content of ANM was measured by high-performance broth chromatography (HPLC). The chromatographic column was an Apollo C18 column; the elution method was CH_3_CN/water mixed solvent gradient elution and the CH_3_CN concentration changed from 10% to 100% in 0–22 min; the flow rate was 1 mL/min; the injection volume was 20 μL; the detection wavelength was 224 nm. After the fermentation, 1 mL of the fermentation broth was centrifuged at 12,000 r/min for 10 min and the supernatant was aspirated and filtered with a 0.22 μM microporous membrane and then directly detected by HPLC. The yield of ANM was calculated according to the ANM standard curve.

### 4.8. Separation Process Optimization

The ANM separation process was explored considering the production cost. Likewise, the 20-ton fermenter was mainly used as the research object. It mainly explored the pretreatment of fermentation broth, the selection of adsorption resin, the selection of analytical broth, the selection of concentration temperature and the scheme of crystallization and drying. In addition, the detection of ANM content also adopts the HPLC method, which is the same as that described in 4.6.

### 4.9. ANM A Biological Activities Test

#### 4.9.1. The Preventive Effect of ANM A on Rice Bacterial Leaf Blight Caused by *X. oryzae* pv. *oryzae* (Greenhouse)

The cutting leaf method: rice plants with 5–6 leaves were inoculated with the pathogen using leaf clipping [38]. First, dip the tip of the scissors into the suspension of *X. oryzae* pv. *oryzae*, the bacteria pathogen of rice bacterial leaf blight and cut along the tip of the rice leaf at 2–3 cm. The method of inoculating pathogenic bacteria by the piercing leaf was used to observe the state of leaves. The main operation method was to pierce 5 small holes on each side of the leaf using a sterile needle tube immersed in the suspension of pathogenic bacteria. The pathogen was previously grown on ub-potato extract agar consisting of sucrose 1.5%, Ca (NO_3_) 2 0.05% and Na_2_HPO_4_ 0.2% [39]. Cultivation at 28–30 °C for 24 h was performed in the room with 100% relative humidity and then at 28 °C for 14 days in the greenhouse. Bis-ADTA was used as the control agent, with each treatment setting 3 replicates. All of the leaves were examined for the presence or absence of disease to calculate the percentage of diseased leaves [40].

#### 4.9.2. The Therapeutic Effect of ANM A on Citrus Canker Caused by *X. citri* (Field)

An experiment on the therapeutic effect of ANM A on citrus canker was carried out in an orange orchard in Tianbian Town, Nanhai County, Guangdong Province. After investigating the disease index (Appendix A), ANM A was formulated into different concentrations and the area with the uniform disease was selected and sprayed evenly on the front and back of the leaves. Next, 14% copper humic acid was diluted 300-times as the control agent and water was the blank control. Each treatment group was repeated 3 times and the results were observed 30 days later.

The pathological grading criteria of disease index were as follows: Grade 0, no extended lesion; Grade 1, the lesions on the leaves did not spread obviously; Grade 3, the area of the lesions was less than one-third of the leaf; Grade 5, the area of lesions was less than two-thirds of the leaf; Grade 7, the area of lesions was less than four-fifths of the leaf, which turned yellow obviously; Grade 9, the lesions corrode the whole leaf, which was withered and dead.

The disease index (DI) = (∑(The number of diseased leaves × The corresponding pathological grade) × 100)/(Total number of test leaves × 9).

### 4.10. ANM (A + B) Biological Activities Test

#### 4.10.1. The Preventive and Therapeutic Effect of ANM (A + B) on Rice Bacterial Leaf Blight Caused by *X. oryzae* pv. *oryzae* (Field)

The protective effect of leaf blight disease was tested in fields in Songjiang District, Shanghai, China. ANM (A + B) was diluted to predetermined concentrations and the leaves were sprayed evenly at the two leaves’ terminal bud stage of rice inoculum lings. After the liquid dried, rice leaves were inoculated with an equal amount of *X. oryzae* pv. *oryzae*. To test the therapeutic effect of leaf blight, the pathogen was inoculated at the booting stage of rice. ANM (A + B) was diluted to a predetermined concentration and then uniformly sprayed on the front and back of the leaves in the area with uniform incidence. During the test period, the water consumption per hectare was 50 kg. Bis-ADTA was used as a control agent and water was used as a blank control. Each treatment group was re-peated 3 times and the results were observed 14 days later.

#### 4.10.2. The Therapeutic Effect of ANM (A + B) on Citrus Canker Caused by *X. citri* (Field)

An experiment on the therapeutic effect of ANM (A + B) on citrus canker was carried out in an orange orchard in Tianbian Town, Nanhai County, Guangdong Province. ANM (A + B) was diluted to a predetermined concentration and then the area with uniform incidence was selected to spray evenly on both sides of the leaves. During the test period, the water consumption per hectare was 50 kg. Kocide was used as a control agent and water was used as a blank control. Each treatment group was repeated 3 times and the results were observed 14 days later.

#### 4.10.3. The Therapeutic Effect of ANM (A + B) on Rice Bacterial Leaf Streak Caused by *X. oryzae* pv. *oryzicola* (Field)

The therapeutic effect of rice bacterial leaf streak was tested in the fields of Hunan and Jiangxi provinces in China. ANM (A + B) was diluted to a predetermined concentration and then the area with uniform incidence was selected to spray evenly on both sides of the leaves. During the test period, the water consumption per hectare was 50 kg. Bismerthiazol was used as the control agent and water was used as the blank control. Three replicates were set for each treatment and the results were observed after 10 days.

### 4.11. Statistical Analysis

At least three independent tests were performed for each analysis. The results showed mean ± SD. Dunnett’s test was performed using an analysis of variance. SPSS 26.0 software was used for statistical analysis.

## 5. Conclusions

The strain SPRI-371 was identified as *S. aureus* by culture characteristics and 16 s rRNA analysis. In order to explore the best conditions for its industrial production, its fermentation process and separation method were specifically studied. In the exploration of the fermentation process, a large 20 m^3^ fermentation tank was used for the experiment. In the test, the best ANM yield was found when the rotation speed was 170 rpm, the pressure was 0.05 MPa and the inoculation age was 36–40 h. In the exploration of dissolved oxygen content, it was found that the dissolved oxygen level remained at 1–30% within 40–80 h and at >60% in the later period resulted in an ANM yield of >3700 mg/L.

In the exploration of ANM process for the separation of fermentation broth, sulfuric acid solution pretreatment was selected and 4# resin was used for adsorption. After adsorption was complete, a 20% ethanol solution was selected for analysis and concentrated under vacuum (60–65 °C) for 4.8 h. Through the above methods, the ANM collection rate was >90%.

The activity of ANM was determined and the results showed that it had an excellent therapeutic effect on three pathogens, including *X. oryzae* pv. *oryzae*, *X. citri* and *X. oryzae* pv. *oryzicola*. In the test, ANM A can play an excellent therapeutic effect when used alone. When ANM A 150 gai/ha + ANM B 75 gai/ha was combined, the treatment efficacy for all three diseases exceeded 85%. In addition, the dosage of ANM was lower than that of commercial drugs and the industrial production cost was lower than that of commercial drugs. Therefore, ANM (A + B) has good development prospects as a candidate fungicide.

## Figures and Tables

**Figure 1 molecules-27-05041-f001:**
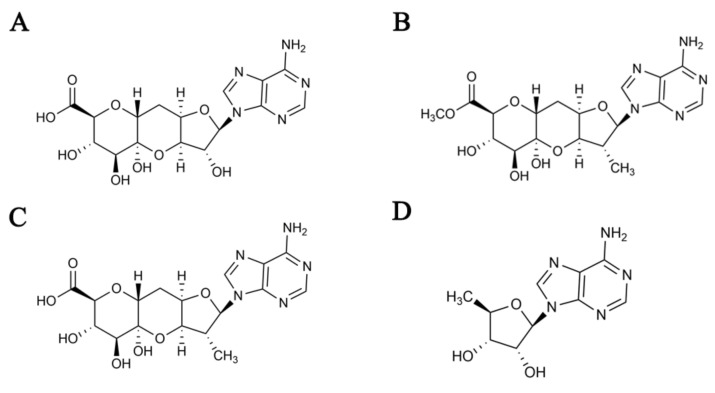
The structure of anreonucleomycin and hepcidin. (**A**) Aureonuclemycin A; (**B**) Aureonuclemycin B; (**C**) Herbicidin C; (**D**) Herbicidin D.

**Figure 2 molecules-27-05041-f002:**
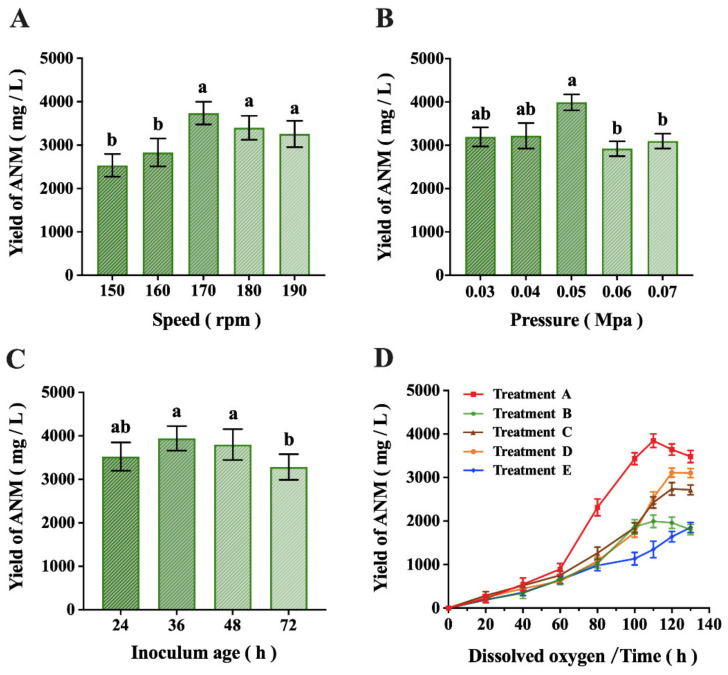
Results of industrial fermentation of SPRI-371 by speed, pressure, inoculum age and dissolved oxygen. (**A**) Speed, rpm; (**B**) pressure, Mpa; (**C**) inoculum age, h; (**D**) dissolved oxygen/time, h. The data are shown as means ± SD of three independent experiments. Lowercase letters indicate significant differences (*p* < 0.05) between any two groups.

**Figure 3 molecules-27-05041-f003:**
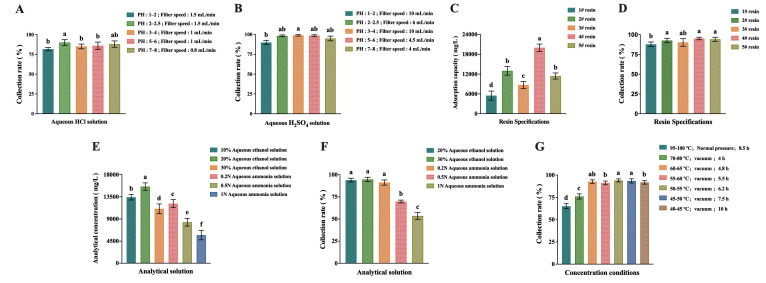
Results of separation process optimization. (**A**) Pretreatment with aqueous hydrochloric acid; (**B**) pretreatment with aqueous sulfuric acid; (**C**) adsorption capacity using different specifications of resin; (**D**) collection rates using different grades of resin; (**E**) concentrations of different analytical solutions; (**F**) collection rates using different analyte solutions; **(G**) collection rates using different concentration temperatures. The data are shown as means ± SD of three independent experiments. Lowercase letters indicate significant differences (*p* < 0.05) between any two groups.

**Figure 4 molecules-27-05041-f004:**
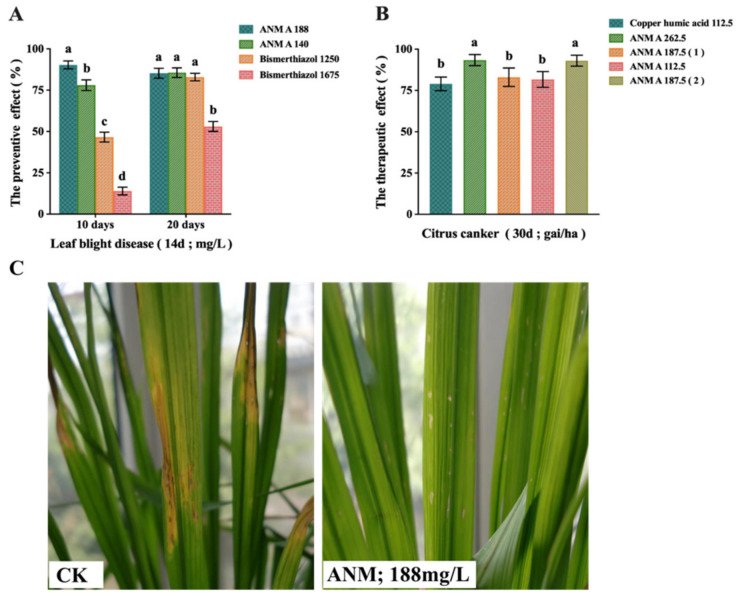
Evaluation of in vivo antibacterial activity of ANM A. (**A**) The cutting leaf method: preventive effect of ANM A on rice bacterial leaf blight caused by *X. oryzae* pv. *oryzae*; (**B**) therapeutic effect of ANM A on citrus canker caused by *X.citri.* (**C**) The piercing leaf method: preventive effect of ANM A on rice bacterial leaf blight caused by *X. oryzae* pv. *oryzae*. The data are shown as means ± SD of three independent experiments. Lowercase letters indicate significant differences (*p* < 0.05) between any two groups.

**Figure 5 molecules-27-05041-f005:**
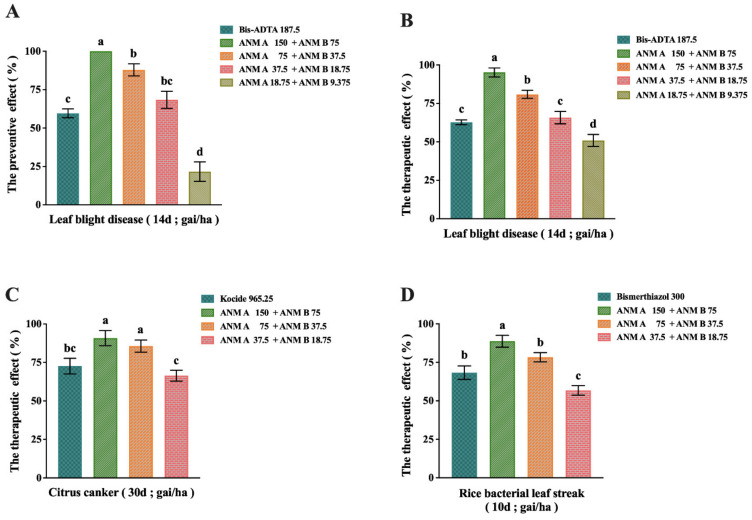
Evaluation of in vivo antibacterial activity of ANM A. (**A**) Preventive effect of ANM (A + B) on rice bacterial leaf blight caused by *X. oryzae* pv. *oryzae*; (**B**) therapeutic effect of ANM (A + B) on rice bacterial leaf blight caused by *X. oryzae* pv. *oryzae*; (**C**) therapeutic effect of ANM (A + B) on citrus canker caused by *X. citri*; (**D**) therapeutic effect of ANM (A + B) on rice bacterial leaf streak caused by *X. oryzae* pv. *oryzicola.* The data are shown as means ± SD of three independent experiments. Lowercase letters indicate significant differences (*p* < 0.05) between any two groups.

**Table 1 molecules-27-05041-t001:** The cultural characters of *Streptomyce**s*
*aureus* strain SPRI-371.

Medium	Growth	Aerial Mycelium	Reverse	Soluble Pigment
Sucrose-nitrate agar	Poor	Scant, white	Yellowish	None
Asparagine-glucose agar ISP5	Abundant	Good, grayish	White	None
Gao’s	Abundant	Good, grayish	White	Slight yellow
Starch agar ISP4	Good	Good, white	Milky	Slight yellow
Tyrosine agar ISP7	Abundant	Good, gray	Brown	Yellow brown
Malic calcium agar	Abundant	Good, grayish	Brown	Brown yellow
Yeast-glucose agar	Good	Good, white	Deep brown	Deep brown
Glycine-asparagine agar	Abundant	Good, gray, brown	Cuticolor, brown	Brown yellow
Oatmeal agar ISP3	Poor	Poor, grayish	Yellow, brown	None
Ke’s	Abundant	Abunt, gray, crapy	White	Slight yellow
Potato	Abundant	Good, grayish,	Brown	Deep brown

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
