# Peer review of "Antibacterial Activity of Aureonuclemycin Produced by *Streptomyces aureus* Strain SPRI-371"

_molecules, 2022, doi:10.3390/molecules27155041_

Round 1
Reviewer 1 Report
The study is a significant contribution to this specific area and it is complete. Nevertheless, the manuscript contains serious agronomic (from the point of view of plant pathology) errors. All this was indicated in the attached pdf with the “comments” function.
One of the main points is that it is not possible to correctly identify the species within the genus Streptomyces with only the partial 16s rRNA sequence. For this it is necessary to carry out a multilocus sequence analysis scheme developed to address a phylogenetic analysis. Therefore, either the authors add this study or, with the information available, they should correct the entire manuscript as "Streptomyces spp"
It would have been nice to see some photos of the plants treated with ANM (in vivo experiments)

Author Response
Dear Reviewer:
On behalf of my co-authors, I’m submitting the enclosed manuscript entitled “Aureonuclemycin with antibacterial activity produced by Streptomyces aureus strain SPRI-371” for possible publication in MOLECULES. The article had been revised by my native English speaking colleagues.Thank you for your comments on this manuscript. Corrections have been made regarding formatting issues in the manuscript.
On the taxonomic study, we agree with the reviewers. 16s rRNA analysis alone cannot classify bacteria into species. Although in the existing studies, this method has been applied to classify, such as the identification of anaerobic bacteria(Justesen,2018). In our study, 16s rRNA analysis was an auxiliary method in order to identify the closest relatives of the species from the Neighbor-Joining tree. In our research, the classification method was still based on the classic Actinomycetes Classification Manual.And The strain SPRI-371 (Streptomyces aureuse var. suzhoueusis n. var. Yan et al) was named with the help of Mr. Xunchu Yan.
In the test of antimicrobial activity against leaf blight disease, we did two inoculation methods of the pathogenic bacteria, including the cutting leaf method and the piercing leaf method. The preventive effect of the two methods of inoculating pathogenic bacteria is basically the same. Photos of leaves after ANM treating have been added to the new manuscript.The new manuscript has been placed in the attachment.
Best Regards.
Yours Sincerely
Yang Zhang

Reviewer 2 Report
In this paper the authors analyzed the antibacterial properties of a molecule produced by the Streptomyces aureus SPRI-371 strain.
Despite the great work done, there are still a lot of spelling, grammar and formal errors. Several sentences must be rephrased; genus and species must always be written in Italic
I strongly encourage the Authors to review the scientific language used in the manuscript by an English mothertongue. In a number of cases, these errors led to misinterpretation of what they want to demonstrate
In general, there is little attention to details by the Authors.
References format must to be uniformed (ex. refs 19, 20, 26, 29 and 30)
Conclusions must be improved. They actually seem rather a summary.
Author Response
Dear Reviewer:
On behalf of my co-authors, I’m submitting the enclosed manuscript entitled “Aureonuclemycin with antibacterial activity produced by Streptomyces aureus strain SPRI-371” for possible publication in MOLECULES. The article had been revised by my native English speaking colleagues.
Thank you for your comments on this manuscript. Corrections have been made regarding formatting issues in the manuscript.The conclusions in the article have been rewritten. The new manuscript has been placed in the attachment.
Best Regards.Yours Sincerely
Yang Zhang
